# Chromatin priming elements direct tissue-specific gene activity before hematopoietic specification

Alexander Maytum*, Benjamin Edginton-White* , Peter Keane , Peter N Cockerill , Jean-Baptiste Cazier , Constanze Bonifer

**Tissue-specific gene regulation during development involves the interplay between transcription factors and epigenetic regulators binding to enhancer and promoter elements. The pattern of active enhancers defines the cellular differentiation state. However, developmental gene activation involves a previous step called chromatin priming which is not fully understood. We recently developed a genome-wide functional assay that allowed us to functionally identify enhancer elements integrated in chromatin regulating five stages spanning the in vitro differentiation of embryonic stem cells to blood. We also measured global chromatin accessibility, histone modifications, and transcription factor binding. The integration of these data identified and characterised cis-regulatory elements which become activated before the onset of gene expression, some of which are primed in a signalling-dependent fashion. Deletion of such a priming element leads to a delay in the up-regulation of its associated gene in development. Our work uncovers the details of a complex network of regulatory interactions with the dynamics of early chromatin opening being at the heart of dynamic tissue-specific gene expression control.**

## Introduction

The development of multicellular organisms requires the activation of different gene batteries which specify the identity of each individual cell type. Such shifts in cellular identity are driven by shifts in the gene regulatory network (GRN) consisting of transcription factors (TFs) binding to the enhancers and promoters of their target genes resulting in the alteration of gene expression (Davidson et al, 2002). Such genes may again code for transcription factors, and thus, a GRN comprises the sum of these regulatory interactions. Importantly, a GRN can be inferred from multi-omics data including chromatin immunoprecipitation (ChIP) assays, global gene expression, and chromatin accessibility which together reveal the TF

motifs underlying gene regulation (Goode et al, 2016; Assi et al, 2019; Bravo Gonzalez-Blas et al, 2023). During development, GRN shifts are initiated by the cellular signalling environment, arising from growth factors, cell–cell contacts and mechanical cues (reviewed in Zaret [2020] and Edginton-White and Bonifer [2022]) which activate intracellular transduction cascades that eventually change gene expression by regulating the activity of signalling-responsive transcription factors. However, development and GRN shifts are also a highly regulated multi-step processes that involve the generation of precursor cell types that further diversify until terminal differentiation is reached. In a similar vein, the actual activation of mRNA synthesis is not the first step in the activation of tissue-specific expressed genes. The release of RNA polymerase II from promoters is a multistep process that requires the coordination of signalling processes, TF binding to enhancers and promoters, and the reorganisation and modification of chromatin to allow the assembly of multiprotein complexes and the interaction of these cis-regulatory elements in intranuclear space (Jonkers & Lis, 2015). It is now well established that the chromatin structure of tissue-specific expressed genes must first be rendered more accessible to allow full gene activation to occur in a process called chromatin priming (Hu et al, 1997; Bonifer & Cockerill, 2017).

Chromatin priming at specific cis-regulatory elements was first observed in genes specific for terminally differentiated cells which are not expressed in multipotent blood progenitors (reviewed in Bonifer and Cockerill [2017]). Moreover, the opening up of chromatin in the absence of gene expression can serve as a molecular memory of a previous signalling event, allowing cells to reactivate genes more rapidly when receiving a second signal (Bevington et al, 2016, 2017). Although the precise relationship between enhancer elements and primed elements and whether they differ in their molecular features remain unclear, at least a subset of priming elements are devoid of enhancer activity when studied in isolation in reporter assays (Bevington et al, 2016; Bonifer & Cockerill, 2017).

We recently developed a global functional method that allowed us to assay enhancer activity of individual elements at five different stages during the in vitro differentiation of embryonic stem

---

Institute for Cancer and Genomic Sciences, College of Medical and Dental Sciences, University of Birmingham, Birmingham, UK

Correspondence: c.bonifer@Bham.ac.uk
*Alexander Maytum and Benjamin Edginton-White are joint first authorship

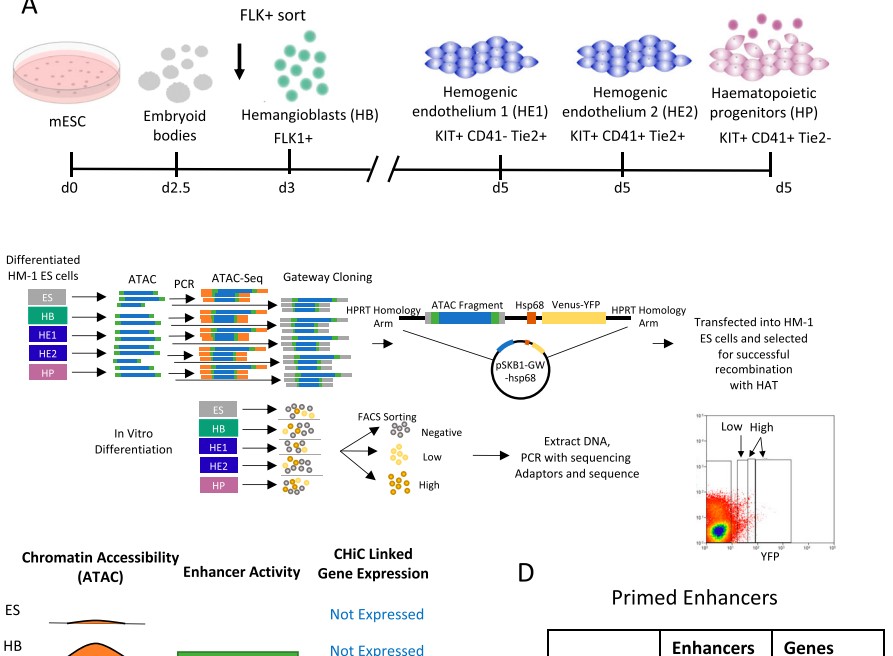

**Figure 1. Identification of primed enhancer elements.**
**(A)** Schematic of the in vitro ES cell differentiation system in serum and the cell types sorted for analysis. **(A, B)** Depiction of the screening method using ATAC-seq fragments from the five cell stages (embryonic stem cells, HB, HE1, HE2, and HP) shown in (A). Bottom lower panel: FACS profile showing how high and low enhancer activity was defined. **(C)** Diagrammatic representation of the chromatin state at a hypothetical cis-regulatory element becoming accessible at the HB cell stage before the onset of gene expression of the associated gene at the HE1 stage. **(D)** Table of the number of cis-elements classified as primed for each of the changes of cell stages and the number of genes primed enhancers are associated with. ES > HB: elements with open chromatin and enhancer activity before the expression of their associated genes in HB. HB > HE and HE > HP: The same for transitions at later developmental stages. **(E)** Heatmap showing the median Z-score of gene expression values for genes associated to primed enhancers for each cell stage transition. **(F)** Percentage of enhancer elements primed at the first stage of each transition that show enhancer activity at later stages.

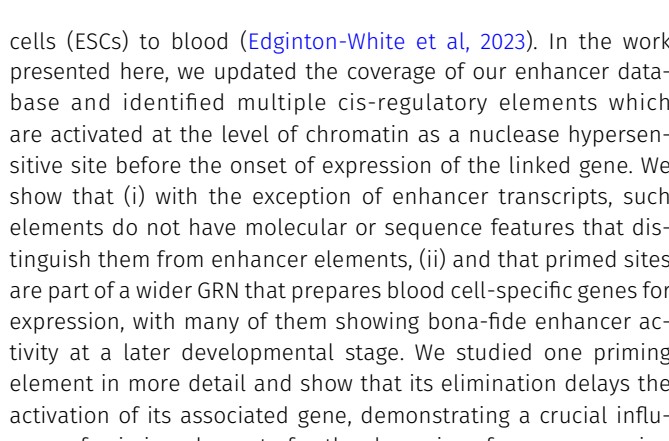

Primed Enhancers

|  | Enhancers | Genes |
|---|---|---|
| **ES>HB** | 330 | 221 |
| **HB>HE** | 294 | 174 |
| **HE>HP** | 237 | 137 |

cells (ESCs) to blood (Edginton-White et al, 2023). In the work presented here, we updated the coverage of our enhancer database and identified multiple cis-regulatory elements which are activated at the level of chromatin as a nuclease hypersensitive site before the onset of expression of the linked gene. We show that (i) with the exception of enhancer transcripts, such elements do not have molecular or sequence features that distinguish them from enhancer elements, (ii) and that primed sites are part of a wider GRN that prepares blood cell-specific genes for expression, with many of them showing bona-fide enhancer activity at a later developmental stage. We studied one priming element in more detail and show that its elimination delays the activation of its associated gene, demonstrating a crucial influence of priming elements for the dynamics of gene expression control in development.

# Results

## Primed enhancers are part of large GRNs of developmentally regulated genes that only become expressed at a later differentiation stage

In our previous work, we used a mouse ESC-based in vitro differentiation system spanning five different stages along the path to blood cell specification (Goode et al, 2016; Obier et al, 2016; Edginton-White et al, 2023) (depicted in Fig 1A): (1) ESCs; (2) FLK1+ hemangioblast cells (HB) which give rise to hematopoietic and endothelial cells; (3) the first stage of the hemogenic endothelium (HE1) which expresses a low level of the master regulator of hematopoiesis, RUNX1; (4) the second stage of the hemogenic endothelium (HE2) where RUNX1 is up-regulated and cells show full

hematopoietic commitment; and (5) the final stage, hematopoietic progenitors (HP) which bud off from the HE2 and are capable of forming mature blood cells. To identify functional enhancer elements, we isolated genomic fragments from open chromatin regions and reintegrated them into the genome at a safe habour reporter site embedded in the ES cell genome as depicted schematically in Fig 1B. These cells were then sequentially induced to differentiate, and cells were purified from the various stages using the surface markers depicted in Fig 1A. Cells harboring active enhancers were purified as YFP+ cells by flow cytometry and the active integrated fragments were identified using next generation sequencing. For this work, we also resequenced the original transfection library to increase the number and coverage of elements (for details see the Materials and Methods section, data listed in Supplemental Data 1 and Supplemental Data 2). After filtering (Fig S1A), our method identified thousands of open chromatin regions that were capable of driving reporter gene expression in a cell type-specific fashion, many of which were distal elements, but also many promoters as previously discussed (Fig S1B and C) (Maytum et al, 2023).

In this work, we were interested in identifying regulatory elements which exist in open chromatin before the onset of expression of their linked gene (a hypothical example of priming in HB is depicted in Fig 1C). Most of these elements also showed enhancer activity in our assay. Because our initial analyses found little difference in stage-specific priming between HE1 and HE2, we only used the HE1 dataset as HE (hemogenic endothelium) for all subsequent analyses. These analyses identified (i) 330 enhancers that were active in ES linked to 221 genes that are activated later in HB (ES > HB), (ii) 294 enhancers that were active in HB linked to 174 genes that are activated later in HE (ES > HB), and (iii) 237 enhancers that were active in HE linked to 137 genes that are activated later in HP (HE > HP) (Fig 1D). To show the overall trend of changes in transcriptional activity of these primed genes we also calculated the Z-score for the mRNA values for each subset and each of the four stages (Fig 1E). We subdivided distal sites into those with low and high enhancer activity as measured by flow cytometry (Fig 1B, bottom right panel, Fig S1D). Additional analyses of enhancer activity in primed genes at the different stages of cell differentiation (Fig S1E and F) demonstrated a highly dynamic behaviour of the different elements. Roughly half of the priming enhancers identified maintained a similar level of enhancer activity at the next stage (High > High or Low > Low), about a quarter increased in activity (Low > High), whereas the remainder decreasing in activity (High > Low). Furthermore, many primed enhancers were active in more than one stage before the associated gene was expressed and/or remained active in more than one stage after activation (Fig 1F). This was particularly pronounced at the ES cell stage where 60% of all elements primed in ESCs stayed active at the HP stage, 80% of all elements primed at the HB stage stayed active in HP cells, and 100% of all elements primed at the HE stage showed enhancer activity at the HP stage. Although a subset of the ESC and HB active enhancer patterns was lost in HP, the hematopoietic HP program was already fully primed at the HE stage and thereby ready to go after the endothelial–haematopoietic transition.

We next asked the question of (i) which genes were associated with primed enhancers, and (ii) how these genes were organised within a wider network of expressed TF genes. To this end, we identified genes inactive at one developmental stage that were associated with primed enhancers and linked them to expressed TFs via their binding motifs as described in Fig 2A top panel. We also identified TF encoding gene loci that harbored primed enhancers (depicted in blue) containing binding motifs for other TFs (incoming arrows) (Fig 2B and C). For the next differentiation stage, when these genes become active, we again linked their previously primed enhancers to the same TFs to examine, how the used motifs had changed (incoming links) and how the activation of expression of these TFs changed outgoing links (Fig 2A, bottom panel). We performed this analysis for the ES to HB (Fig 2B), the HB to HE (Fig S2A), and the HE to HP transitions (Fig S2B). The results of these analyses provide a detailed picture of how one developmental stage of cell specification is anticipated in the previous developmental stage. In ES cells, the pluripotency signature was obvious with multiple motifs of OCT and SOX factors, but not NANOG, associated with primed enhancers. Moreover, ES cells contained the largest numbers of primed enhancers linked to TF encoding genes (Figs 2B and S3A). Differentially regulated genes associated with enhancer priming often contained more than one primed enhancer (Fig S3B) and were mostly associated with important developmental processes (Fig S3C).

Our analysis shows that motif use of primed enhancers within one gene locus changed during differentiation. For example, the *RUNX1* locus (indicated by an arrow) was not expressed in ES cells, but was already primed by a cis-element containing a SMAD motif (note the arrow from SMAD to *RUNX1*). At the HB stage, *Runx1* is barely expressed, but is now linked to genes activated by elements containing RUNX1 motifs. At the HB to HE and HE to HP transitions, *Runx1* is linked to different primed genes which then become activated, some of them, such as *Gfi1*, by the direct action of RUNX1 (Lancrin et al, 2012). ETS factors, including ETV2, which has been shown to bind to hematopoietic genes before lineage commitment (Steimle et al, 2023), are widely used at each developmental stage. The gene encoding another important hematopoietic TF, TAL1, is primed in ESCs and then becomes up-regulated and associated with enhancers at the HB stage as also shown by ChIP (Goode et al, 2016). Towards the end of blood specification, the narrowing of developmental potential at the HE stage is associated with a near absence of primed TF genes, the extensive use of TFs known to drive hematopoiesis and the loss of links from specific TF families, such as TEAD, indicating that the enhancer code has changed and the respective motifs are not used anymore.

Taken together, our analyses show in fine detail how the opening up of chromatin anticipates the further use of TFs expressed later in development. To be able to visualise the different connections, we have created a web resource that makes it possible to see the different connections within a high-resolution image. The link is highlighted in the figure legend and the Materials and Methods section.

## Molecular features of primed enhancer elements

We next investigated whether primed enhancers show distinct features as compared with active enhancer elements associated with active genes. It has been reported that active enhancers bind

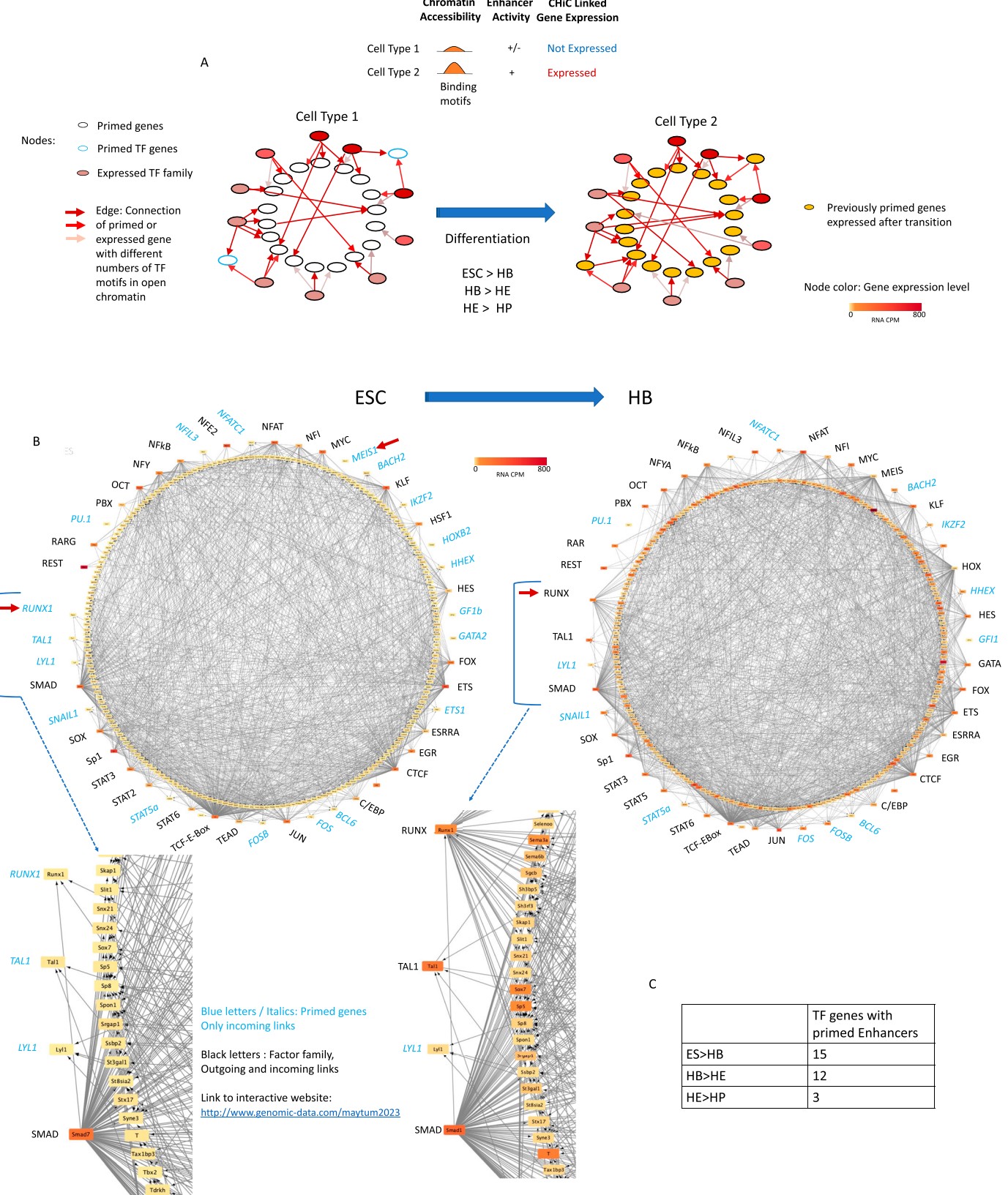

RNA polymerase II and are transcribed to make eRNAs (Lam et al, 2014). To examine whether this was also true for primed enhancer elements, we measured nuclear RNA by RNA-seq and correlated the presence of RNA sequences at intragenic sites which make up about half of all primed sites with the primed state (Fig 3A). We found a strong correlation for the presence of eRNAs at enhancers associated with active, but not with inactive genes, indicating that eRNA transcription is not associated with enhancer priming but gene activation. An example for such a non-transcribed primed element at the *Bmper* gene is shown in Fig 3B. This gene contains an enhancer that is primed in ES cells and becomes active in HB where gene activation is associated with a burst of noncoding transcription on the reverse strand.

In our previous work, we used ChIP assays to measure TF binding and histone modifications at five stages of ES cell differentiation to blood (Goode et al, 2016; Obier et al, 2016; Gilmour et al, 2018; Kellaway et al, 2021), as listed in Fig 3C. Histone H3 lysine 27 trimethylation (H3K27me3) has previously been associated with poised chromatin waiting to be transcribed, whereas histones flanking active enhancer elements become acetylated (Ac) at H3K27 (Rada-Iglesias et al, 2011). We therefore compared the presence of H3K27me3 and H3K27Ac at primed enhancers associated with inactive genes and enhancers associated with active genes (Fig 3D–F). The average profiles depicted in Fig 3D–F show an overall decrease in H3K27me3 signal as elements transitioned from the primed to the active state. Interestingly, H3K27me3 signals were more highly focussed at primed elements in ES suggesting positioned nucleosomes and were replaced by both H3K27me3 and H3K27Ac flanking an open chromatin site in HB.

We next asked the question of whether there were any features, such a specific combination of TF binding/motifs or histone modifications associated with primed enhancers which distinguished them from enhancers associated with active genes (Fig 3G). To this end, we overlapped our enhancers and primed elements with histone and TF ChIP-seq data and searched for a wide selection of TF motifs in each site. From these overlaps, we produced a binary matrix and plotted the proportions of each feature in the enhancers versus primed elements for each transition as indicated in the figure. With the exception of H3K27me3 in ES cells, no other features distinguished primed enhancers from enhancers associated with active genes.

Taken together, our data show that (i) motif use of primed elements changes during differentiation, as it does for active enhancers; (ii) that eRNA presence is a hallmark of enhancers associated with active genes; and (iii) that the polycomb-associated

H3K27me3 mark is associated with the primed enhancer state only in ESCs but not in differentiating cells.

## VEGF signalling represses enhancer priming at hematopoietic genes

In our previous work, we used chromatin accessibility assays (ATAC-Seq) performed with cells isolated from a serum-free ES cell differentiation system which allowed us to examine the effect of specific cytokines on enhancer activation (Pearson et al, 2008; Edginton-White et al, 2023; Maytum et al, 2023). We found that one cytokine, VEGF, is instrumental in activating enhancers driving the expression of endothelial genes by activating the TF AP-1 which cooperates with the Hippo-signalling component TEAD (Obier et al, 2016; Edginton-White et al, 2023). VEGF signalling also acts to up-regulate NOTCH1 signalling components and directs SOX17 which along with TEAD represses *Runx1*. We also showed that VEGF needs to be withdrawn to enable the activation of hematopoietic enhancers, notably those of *Runx1* which is essential for the expression of blood cell-specific genes such as *Spi1* (PU.1) (Huang et al, 2008). The presence of VEGF blocked the activation of *Runx1* enhancers even when studied in isolation (Edginton-White et al, 2023). We therefore examined our primed enhancer collection in HE cells (237 HE > HP elements, see Fig 1D) to identify those responsive, that is, being present or absent depending on the presence or absence of VEGF (Fig 4A), as outlined in the top panel of Fig 4B. Gene expression was measured by single-cell RNA-seq in HE and HP (all data from Edginton-White et al [2023]). 34 genes contained primed elements which were suppressed by the presence of VEGF and became activated in HE once VEGF was withdrawn and significantly increased gene expression in HP cells (Fig 4B, bottom panel). Examples for such genes and elements are shown in Fig 4C–E, with primed elements being highlighted. The most important gene in this collection is the *Spi1* locus which encodes the master regulator of myelopoiesis, PU.1 (Fig 4C). This gene is a direct target of RUNX1 which binds to an enhancer element at −14 kb upstream of the promoter (3′URE [Huang et al, 2008]). This element is primed in the HE and in the presence of VEGF where the gene is silent/lowly expressed. Once VEGF is removed, RUNX1 is up-regulated and together with PU.1 itself and other factors activates this and other enhancers within the locus and up-regulates PU.1 expression (Lichtinger et al, 2012). Other genes associated with VEGF-responsive enhancers include the genes encoding TFs (TOX, BCL11a), multiple signalling proteins such as the hematopoietic cytokine receptors for CSF3 and CSF2, and cytokines such as SPP1 (osteopontin).

**Figure 2. Primed enhancer elements form large gene regulatory networks (GRNs) that set up the next differentiation steps.**
**(A)** Schematic of GRNs changing through differentiation. **(B)** Schematic of GRNs (priming networks) formed by transcription factor encoding genes forming nodes (coloured circles) when their gene products (TFs) bind to other genes (outgoing arrows) which are not yet expressed, shown here for the transition from embryonic stem cells to HB. The analysis is based on the presence of binding motifs in the target genes, TFs are therefore listed as factor families capable of binding the same motif. The node colour represents the gene expression value of a gene within the given cell type. Targets with incoming arrows are listed as individual genes. Below the figure, we zoomed in on a slice of the network which we describe in the text; definitions are listed in between the slices together with the link to the interactive website (see below). (i) Genes that are not yet expressed (yellow node colour), but primed and thus are already connected to other genes (incoming connections only), are marked in blue and by their gene name in italics (example: *Runx1*), (ii) TF genes that are already expressed (orange–red node colour) and the TFs has a motif on enhancers of other genes (incoming and outgoing connections) are marked in black letters (example: SMAD). The ring of TF encoding genes is linked to a ring of target genes which are first not expressed and then expressed (change of colour from yellow to orange). Details of which TF is associated with which gene can be seen using the following link: http://www.genomic-data.com/maytum2023. **(C)** Number of genes encoding TFs with primed enhancers at each developmental stage.

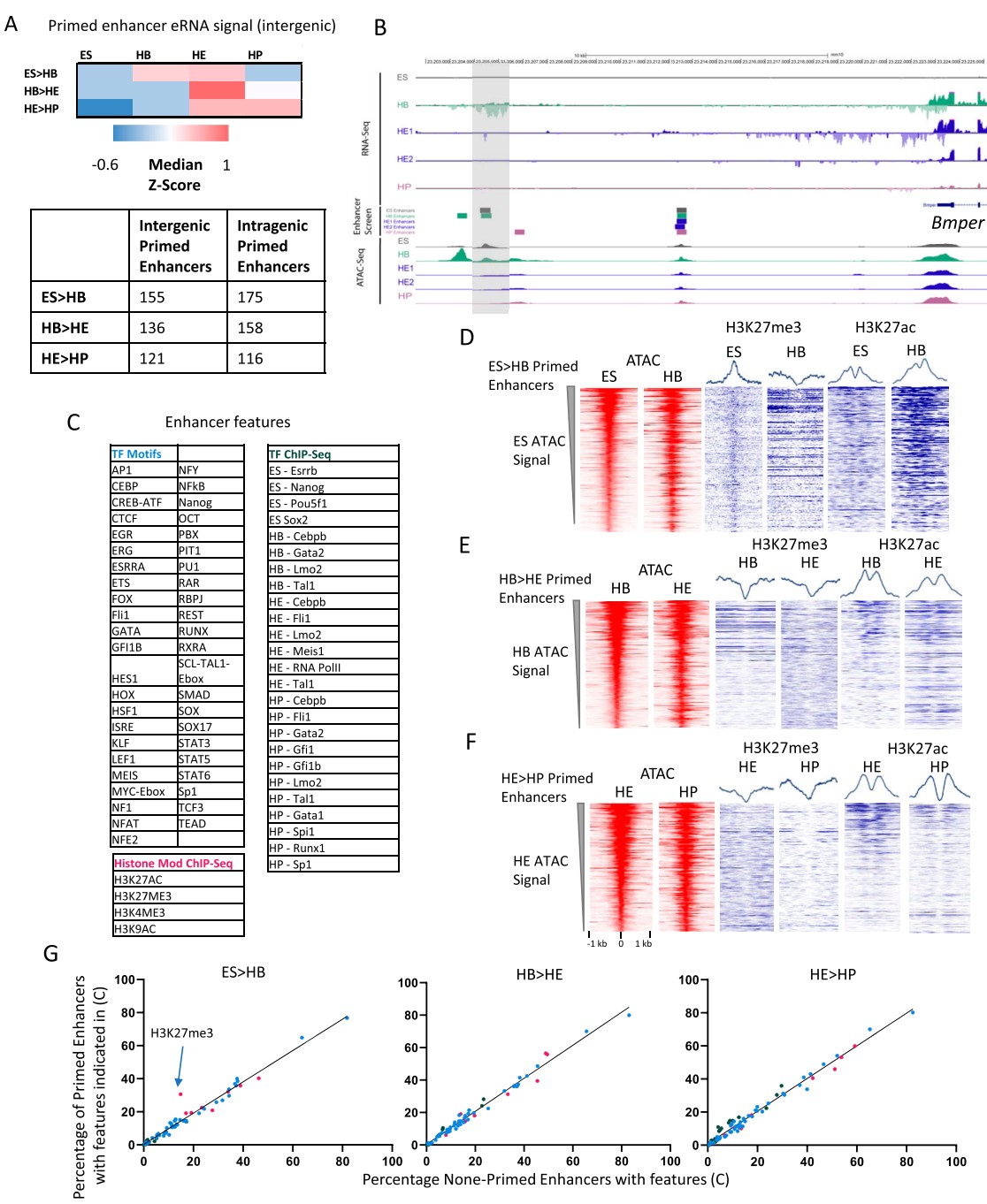

**Figure 3.  Primed enhancer elements are not transcribed but show the same molecular features as enhancers associated with active genes.**
**(A)** Heatmap showing the median Z-score, representing RNA-seq tag counts at each intergenic primed enhancer across each stage of differentiation. This shows the dynamics of eRNA expression from primed enhancers. Bottom panel: table of the number of intergenic and intragenic primed enhancers for each cell stage transition. **(B)** Screenshot showing an enhancer (highlighted by a grey bar) primed in the HB for the *Bmper* locus with eRNA absent when the enhancer is primed and present when the associated gene becomes active. **(C, G)** Table of TF binding and chromatin features at enhancers feeding into the analysis shown in (G). Data from Goode et al (2016), Obier et al (2016), and Gilmour et al (2018). **(D, E, F)** Open chromatin regions and histone modifications (H3K27me3 and HK27Ac) of primed elements throughout the different differentiation stages. For each transition pair, ATAC-seq tag counts at primed sites were ranked. All other data were ranked alongside. The average profiles for each plot are placed on top. **(G)** Comparison of molecular features of priming elements and enhancers during the indicated transitions. **(C)** Plot of proportion of primed elements and active enhancer elements overlapping with features indicated in (C). **(C)** Colours of data points correspond to the type of feature, indicated in (C). Line shows linear regression, *P* < 0.0001, R squared = ES > HB 0.9654, HB > HE 0.9823, HE > HP 0.9791. The H3K27me3 mark in embryonic stem cells is highlighted in the left panel.

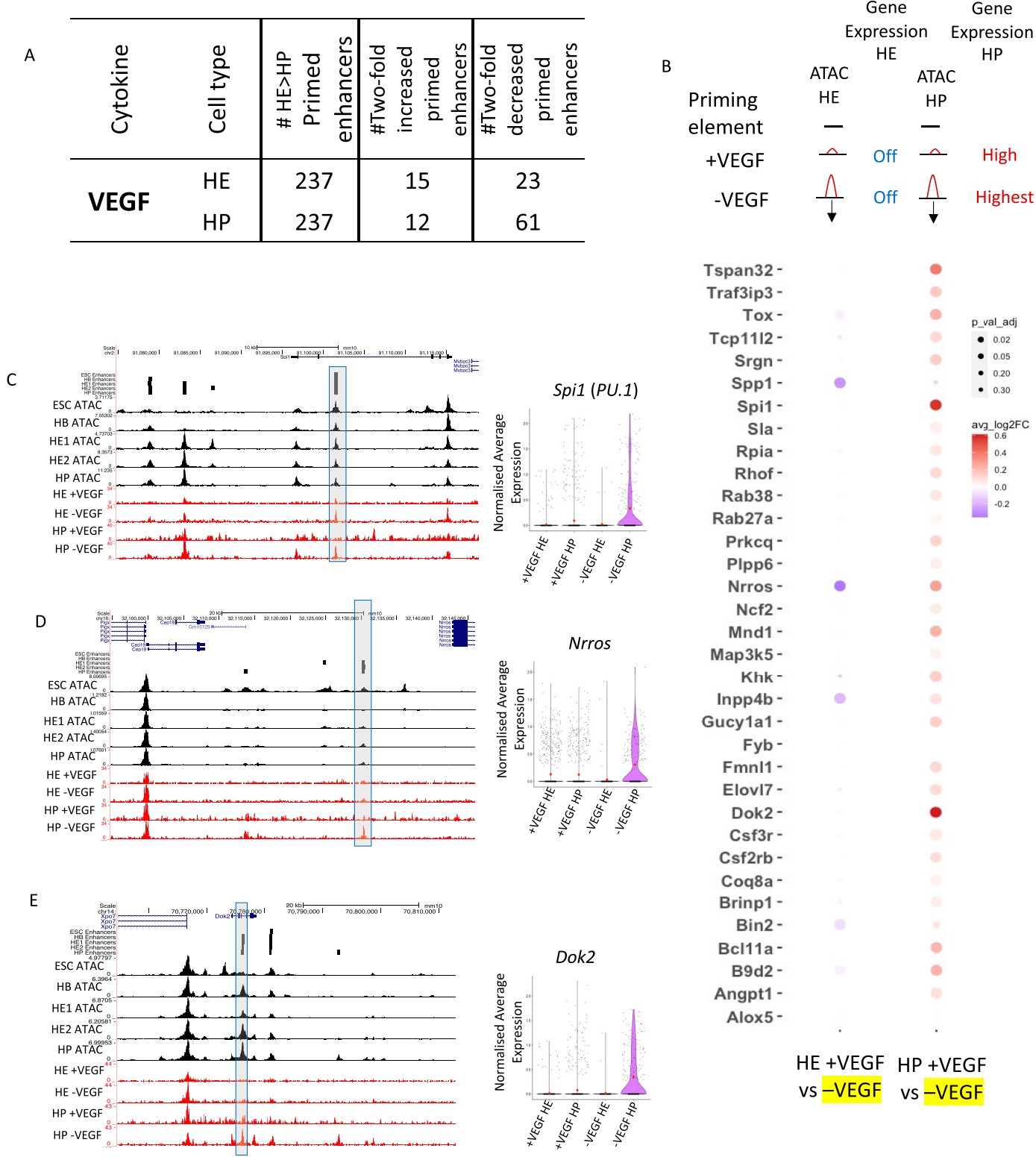

**Figure 4. Primed elements can be established by responding to the signalling environment and can acquire enhancer activity when studied in isolation.**
**(A)** Table showing the number of primed enhancers predicted as being VEGF responsive to in the HE and HP cell types, that is, being present or absent in the presence or absence of VEGF whereby the absence of VEGF primes the hematopoietic fate. **(B)** Dot plot of differential gene expression analysis of genes linked to priming elements which are increased in the absence of VEGF from HE and HP. Dot colour shows the fold change in gene expression and dot size shows the Bonferroni-corrected P-value. Top of panel shows a diagrammatic representation of the chromatin state of the genes associated enhancer elements and the gene expression of the genes in HE and HP. **(C, D, E)** UCSC browser screenshots of ATAC data showing example priming elements for *Spi1* (*PU.1*), *Nrros*, and *Dok2*. The black tracks represent ATAC-seq data from

These experiments demonstrate that extracellular signalling, in this case, VEGF signalling, is truly cell-fate instructive by impacting on primed enhancer elements via modulating the expression and activity of TF genes. Its presence or absence decides the fate of endothelial cells with its absence at the hemogenic endothelium stage being essential for the up-regulation of hematopoietic genes.

## Tissue-specific GRNs regulate the priming and activation of single enhancers

As seen in Fig S1E and F, the fact that a priming element is associated with an inactive gene does not mean that the element is unable to up-regulate the activity of a minimal promoter at this developmental stage, that is, displaying functional enhancer activity. We therefore asked the question of how individual primed enhancer elements operate in isolation and within the context of an entire gene locus. To this end, we first cloned individual priming elements found in HE or HP-specific expressed genes which were found to be organised in open chromatin before the onset of gene expression into our targeting site and measured their activity by flow cytometry (Fig S4A–D). We first investigated priming elements of the *Hand1* cardiovascular gene and the endothelial-specific gene *Kdr* encoding FLK1 (VEGF receptor) which we could link to their respective promoters using CHi-C data (see the Materials and Methods section). Both genes were strongly up-regulated at the HB stage, but the stimulatory activity of their priming elements in isolation at this stage was barely measurable (Fig S4A and B). Moreover, activity of these elements was repressed at later developmental stages (HP) where gene expression went down. A similar pattern was seen at the *Meis1* locus which was up-regulated from the HB stage onwards and contains a primed element at −45 Kb (Fig S4C). Here, priming element activity did not represent the activity of the whole gene locus. A different pattern was seen with the −14 kb *Spi1* 3′URE. Enhancer activity was initially low, but then was strongly up-regulated at the HP stage when RUNX1 and C/EBPα are expressed and bind to this element (Leddin et al, 2011), thus mirroring the activity profile of the whole gene. Another example of a priming element representing the activity of the whole locus is in *Mecom* which encodes the proto-oncogene EVI1 (Figs 5A and S4C) (Ayoub et al, 2018). The gene is up-regulated in HE1 but is then sharply down-regulated (Fig 5A and B). *Mecom* has recently been shown to be essential for the development of endothelial cells (Lv et al, 2023). It contains an enhancer located at +340 kb which is primed in HB but when examined in isolation this element did not show high enhancer activity in our reporter assay until the HE1 state (Fig 5C). Closer inspection of the fragments with enhancer activity showed a shift of the open chromatin region towards the 3′ end of the gene (Figs S5A and B and S6A). The analysis of the motif content showed that this element is bimodal with a 3′ endothelial motif signature (RBPJ and AP-1),

which functions as priming element and acquires HE-specific enhancer activity, and a 5′ hematopoietic signature (RUNX1, ETS, GATA/E-Box [TAL1]) which is associated with enhancer repression (Fig S6B). Closer inspection of its features using published data (Edginton-White et al, 2023) revealed that this enhancer does not only represent a priming element, but is also VEGF responsive (Fig S6A) with an open chromatin region appearing in the HB and the HE under the All cytokine condition, which then becomes more prominent when VEGF is withdrawn but then disappears at the HP stage. The examination of TF-binding data from cultures containing serum (Goode et al, 2016; Obier et al, 2016; Gilmour et al, 2018) showed that the primed enhancer is bound by LMO2 and TAL1 in the HB, whereas the promoter is associated with bivalent chromatin. At the HE stage, when the enhancer is activated, occupancy shifts and the enhancer binds FLI1, LMO2, and TEAD4; thereafter, at the HP state, the promoter histones become acetylated to revert to the bivalent state with loss of an open chromatin at the enhancer.

Taken together, these data confirm that in isolation, priming elements are diverse: they may show little enhancer activity by themselves at a specific stage, but can in some cases acquire high stimulatory activity at later developmental stages by binding additional TFs or by being repressed, indicating that a larger network of factors is modulating their activity. Our data therefore provide a molecular explanation for the cell-stage specific sensitivity of gene expression to perturbation of individual enhancers within individual gene loci (Luo et al, 2023).

To address this issue in more detail in the context of the entire *Mecom* locus, we used CRISPR/Cas9 to remove the +340 kb element and measured gene expression during a time course (16, 40, and 64 h) of blast culture differentiation from FLK1+ HB cells (Fig 5D) together with the expression of CD41 which is a direct RUNX1 target (Fig 5E). The development of the different precursor cell types was measured alongside (Fig 5F). In the HB, the *Mecom* promoter was already active and organised in open chromatin, together with the priming element at +340 kb (Fig 5A). The 5′ end of the locus was marked by a constitutive open chromatin region containing a CTCF motif which is part of the promoter. All other enhancers were still inactive. In the HE stage, multiple enhancers came online which was associated with high-level transcription (Fig 3A). Thereafter, at the HP stage, *Mecom* was repressed and appeared to be reverting to its primed state (Fig 5C). The removal of the +340 kb enhancer led to a strong reduction of gene expression at all developmental stages in spite of the presence of multiple additional enhancer elements which are activated at the level of chromatin at the HE stage. Moreover, we noted a strong reduction in the development of HE cells expressing RUNX1 (CD41+ cells). However, the actual developmental trajectory was not affected as at later stages of differentiation the proportion of cells committed to the hematopoietic fate (HP) within the population slowly catched up once other cis-elements became activated. Over time, a threshold of activation was crossed and the cells

---

cells grown in serum culture, and the red tracks represent cells (HE and HP) in a serum-free medium with and without VEGF. Violin plots showing the normalised average expression of each gene in the presence (+VEGF) and absence (−VEGF) of VEGF for HE and HP based on previously published single-cell data (Edginton-White et al, 2023).

chr3:30353804-30354703

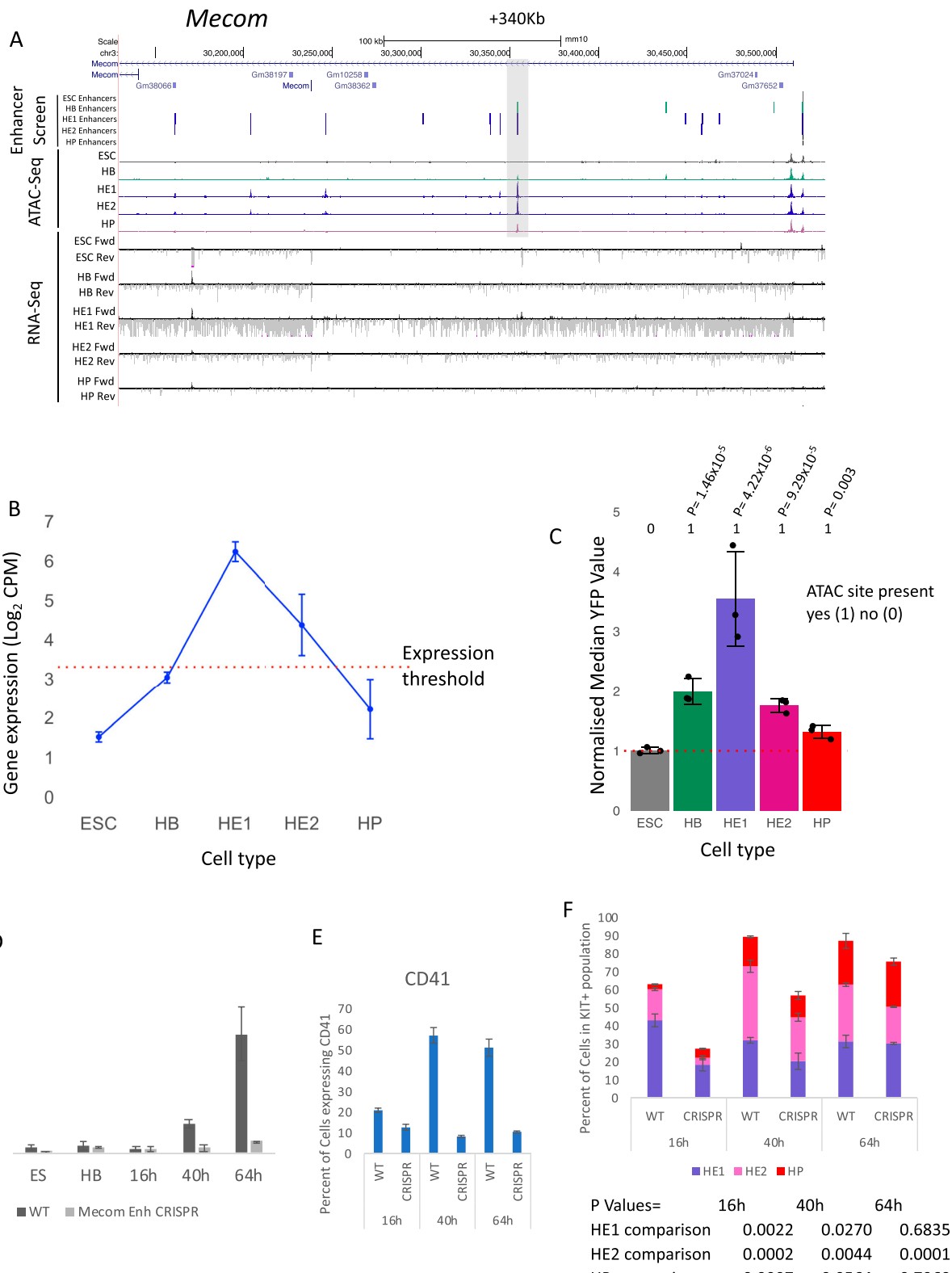

were able to transit into HP cells, most likely because of the accumulation of RUNX1 and other hematopoietic factors in the cell in the absence of cell division. In summary, our data demonstrate the importance of a priming element for correct cell stage-specific gene activation.

# Discussion

Our detailed analysis of the timing of gene activation at the level of chromatin highlights a number of important principles in how cis-regulatory elements activate developmentally controlled genes, summarised in Fig 6. Our updated enhancer database identifies hundreds of enhancer elements which exist as open chromatin regions before the onset of expression of cell stage-specific expressed genes. These primed elements do not yet influence gene expression but already form a vast interconnected network of TF-bound cis-elements that open up the chromatin of precursor cells for the execution of the following cell fate decisions. Primed elements are associated mostly with genes specific for the next developmental stage and the architecture of loci associated with primed elements can involve an already set-up promoter (as in *RUNX1* or *Mecom*) or a closed promoter (as in *Dok2*). When examined in isolation and during differentiation, priming elements do not always have stimulatory activity when assayed in isolation (as in *KDR*), and some are highly active and are then being repressed at a later stage (as in *Mecom*) and others acquiring strong enhancer activity at a late differentiation stage (such as *Spi1*). This complex interplay of different cis-regulatory elements is responsible for the timing of gene activation within a specific GRN.

The work described here also addressed the nature of the molecular difference between priming enhancers and enhancers associated with actively transcribed genes. Comparisons of multiple features of the two enhancer classes at different developmental stages revealed essentially no difference (Fig 3C): All elements were flanked by modified histones and bound TFs specific for the cell state of the respective cells—with one exception: ES cells. In these cells, priming elements were abundant and were associated with nucleosomes positioned over or near the open chromatin region and carrying the H3K27me3 mark. We believe that this feature is a hallmark of ESCs which are kept at an undifferentiated state (Azuara et al, 2006; Rada-Iglesias et al, 2011), whereas cells purified from our in vitro differentiation system represent

a dynamic cell population with rapidly changing cell fates. Moreover, Polycomb group complexes have mostly been associated with promoter elements (Voigt et al, 2013).

We have also shown that in HP cells, 25% of our functionally identified enhancer elements are bound by RNA-polymerase II (Edginton-White et al, 2023). Enhancer transcription has been shown to be important for the activity of entire gene loci (reviewed in Field and Adelman [2020]), most likely because of the fact that it is required for the establishment of enhancer–promoter interaction (Fitz et al, 2020). We did not see enhancer transcription to be globally associated with priming elements as it only seems to be associated with enhancers associated with active genes. However, note that some priming enhancers are paired with an already open, but inactive, promoter which could mean that an interacting complex has already been set up.

We have previously shown that hundreds of enhancer elements respond to extrinsic signals such as cytokines, with the presence or absence of VEGF dictating the choice between endothelial and hematopoietic fate (Edginton-White et al, 2023; Maytum et al, 2023). An important finding from our study was therefore that this feature also holds true for priming elements. We identified priming elements associated with the activation of hematopoietic genes such as *Spi1* (PU.1), *Csf3r* or *BC11a* at the HP stage in the absence of VEGF. An important VEGF-responsive gene is *Mecom* which is transcriptionally activated at the HE stage and is a major regulator for endothelial cell development (Lv et al, 2023). Our work shows that its +340 kb enhancer element which is primed at the HB stage is crucial for the tissue-specific activation of this gene. Therefore, one conclusion of our study is that not only does signalling contribute to regulating the activity of active enhancer elements in development, but also to the set-up of developmental programs that are executed later. It follows that, depending on whether a cell has seen a stimulus, TFs directing a specific cell fate will encounter a different chromatin landscape and the genomic response will be different. We could indeed show that the timing of withdrawal of VEGF is essential for the execution of the endothelial–hematopoietic transition and for the binding of hematopoietic TFs (Edginton-White et al, 2023). Another example of how signalling is used to direct cell fates is seen in the neuronal development of *C. Elegans* where the priming of an early enhancer by Notch signalling leads to a difference in the up-regulation of a mi-RNA, and a difference in left–right asymmetry, depending on whether cells received the signal or not (Cochella & Hobert, 2012). This type of

---

**Figure 5. The removal of a signalling-responsive primed enhancer element at the *Mecom* locus does not alter the differentiating trajectory but reduces the number of hematopoietic cells.**
**(A)** UCSC Browser screenshot showing the *Mecom* locus. The +340-kb priming enhancer is highlighted by a grey box. Distal cis-regulatory elements scoring positive in the enhancer screen are depicted as black vertical bars for each of the five cell types. The ATAC-seq peaks for each of the five cell types shown. Bulk RNA-seq analysis showing the forward (Fwd) and reverse (Rev) reads for each cell type. **(B)** Average gene expression value (Log$_2$ counts per million) of *Mecom* in each of the five cell types. n = 2 biologically independent replicates, whiskers show the value of each replicate, red dotted line shows the threshold value determined as being expressed. **(C)** Activity profile of +340 kb *Mecom* enhancer element in each of the five cell types. Enhancer activity was normalised against the median FITC value of the minimal promoter (MP) control depicted by the red dotted line. n = 3 biologically independent replicates, error bars show the SD, significant *P*-values shown, *P*-values were calculated using two-sided *t* test. Presence or absence of an ATAC site at the five cell stages is expressed as a binary code (1 = scoring positive; 0 = scoring negative). **(D)** *Mecom* gene expression, measured by qPCR, in cells carrying the wild-type locus or a locus with a homozygous deletion of the +340 kb enhancer (CRSPR) in the indicated cell types and over a time course of blast culture (n = 3 biologically independent replicates). **(E)** Expression of CD41 (*Itga2B*) which defines the number of cells passing the RUNX1 threshold allowing cells to conduct the EHT on the surface of cells measured by flow cytometry (n = 3 biologically independent replicates). **(F)** Composition of cell types during the time course of blast culture as measured by flow cytometry (see Fig 1A). n = 3 biologically independent replicates. The *P*-values for the individual differences are listed below the plot, *P*-values were calculated using two-sided *t* test.

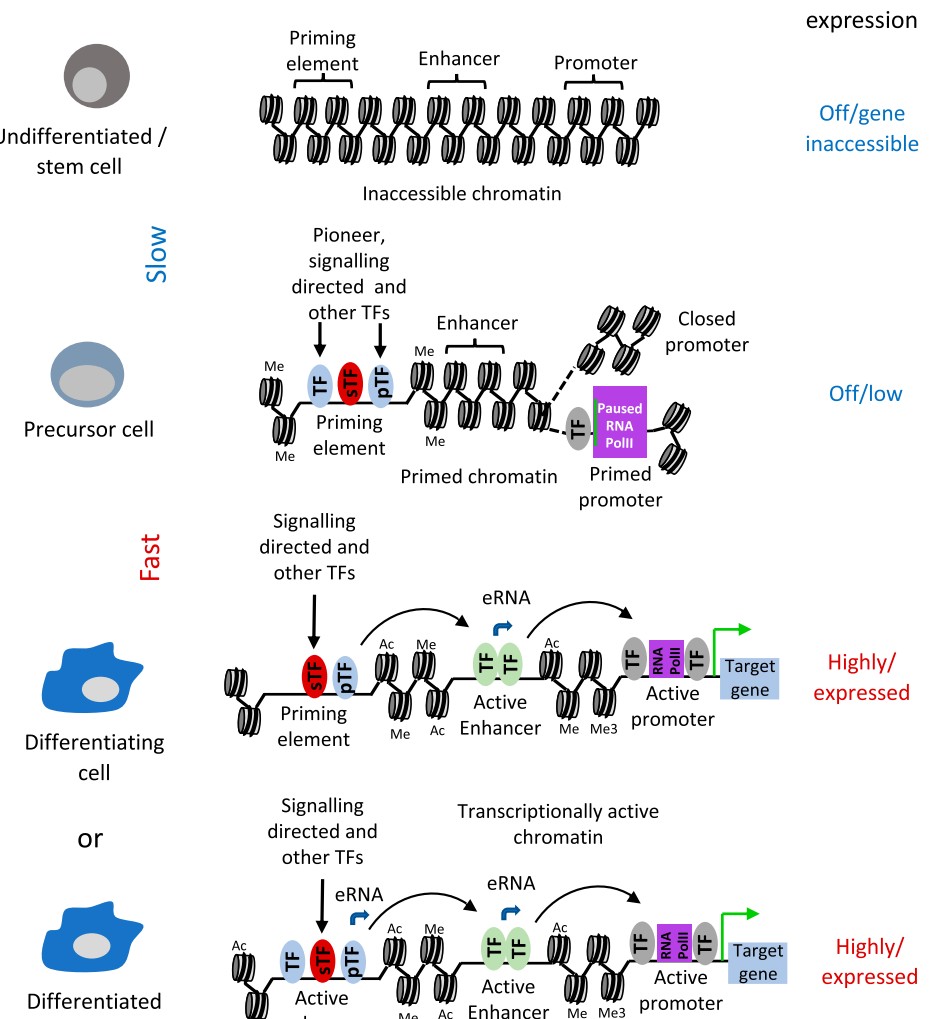

Figure 6.   Model of enhancer priming through cell fate transitions.
Nucleosomes are depicted as round shapes with DNA wrapped around them. Me: H3K4 mono- or tri methylation. Ac: H3 K27 or K9 acetylation. For model interpretation, see the main text.

signalling dependent priming to drive developmental changes differs from that of molecular memory seen in the immune system: Here, signalling activates previously inactive chromatin in naive cells that have not seen a stimulus before. Once the stimulus is gone, cells fall back into a quiescent state but maintain a memory of recent activation ([Bevington et al, 2016; Bonifer & Cockerill, 2017; Pascual-Garcia et al, 2022] and references therein). If a second stimulus arrives, the response is much more rapid but cells maintain their overall identity.

An important feature of priming elements is the fact that deleting them affects the timing of cellular development, as demonstrated by deleting the +340 kb *Mecom* enhancer. Here, the activation of gene expression at the HE1 stage is reduced until the GRN has passed a threshold that brings the other enhancers online and the proportion of HP cells within the population catches up (albeit at reduced numbers). It has been shown that *Mecom* expression is crucial for the expression of VEGFR2 which is a major receptor for VEGF signaling (Lv et al, 2023) thus setting up a feed forward loop that provides a molecular explanation for

this result. A similar phenomenon was seen when the *Spi1* (PU.1) –14 URE was mutated by eliminating a PU.1 autoregulatory binding site; up-regulation of gene expression was delayed until additional enhancers were activated (Lichtinger et al, 2012). These experiments demonstrate that removing or crippling cis-regulatory elements involved in priming from an endogenous gene locus leads to a profound difference in the kinetics of cell fate decisions in development, and it is this feature which is most likely to be affected in elements that are altered by mutations or single-nucleotide polymorphisms scoring in GWAS analyses (Choudhuri et al, 2020). Chromatin priming and the precise timing of the activation of enhancer elements driving gene expression at the right time and in the right cell type is therefore at the very heart of all coordinated, synchronised cell differentiation processes that create fully developed multicellular organisms. Directed cell differentiation needs to take these processes into account. Our priming element resource adds an additional dimension to our ability to interrogate the fine details of hematopoietic specification from ESCs.

# Materials and Methods

### ES cell culture

A HM-1 targeting ES cell line (Magin et al, 1992) was cultured on gelatinised tissue culture plates in DMEM-ES media (DMEM [D5796; Merk], 15% FSC, 1 mM sodium pyruvate [S8636; Merck], 1x penicillin/streptomycin [P4333; Merck], 1 x L-Glutamine [G7513; Merck], 1 x nonessential amino acids [M7145; Merck], 1,000 U/ml ESGROLIF [ESG1107; Merck], 0.15 mM MTG and 25 mM Hepes buffer [H0887; Merck]) at 37°C and 5% $CO_2$. Every 48 h, cell colonies were dissociated with TrypLE Express (12605010; Thermo Fisher Scientific) and replated onto gelatinised tissue culture plates at 1.2 × $10^4$ per $cm^2$.

### In vitro hematopoietic differentiation in serum

Hematopoietic in vitro differentiation (I.V.D) using serum was performed as described in Obier et al (2016) and Edginton-White et al (2023). First, ES cell colonies being cultured on gelatinised plates were dissociated to form a single-cell suspension using TrypLE Express (12605010; Thermo Fisher Scientific). DMEM-ES media were then added at a 1:1 ratio to stop the trypsin activity. The cells were then centrifuged at 300$g$ for 5 min before being resuspended at 2.5 × $10^4$/ml in I.V.D media (IMDM [I3390; Merck], 15% FCS, 1 x penicillin/streptomycin [P4333; Merck], 1 x L-glutamine [G7513; Merck], 0.15 mM MTG, 50 $\mu$g/ml ascorbic acid, and 180 $\mu$g/ml human transferrin [T8158; Merck]) and then plated onto non-adherent dishes (501V; Thermo Fisher Scientific). The cells were then incubated at 37°C, 5% $CO_2$ for 3 d to allow for the formation of floating embryoid bodies (EBs).

The floating EBs were then transferred to 50 ml centrifuge tubes and the EBs were allowed to settle by gravity over 5 min. The EBs were then washed with PBS before being allowed to settle again over 5 min. The EBs were then dissociated with TrypLE Express (12605010; Thermo Fisher Scientific) with gentle pipetting.

The dissociated cells were sorted for the FLK1 surface marker (forming the population referred to as the HB). The dissociated cells were incubated with a FLK1 biotin-coupled antibody (1:200) (13-5821-82; eBioscience), and then with MACS anti-biotin beads (130-090,485; Miltenyi Biotec). FLK1-expressing cells were sorted for by separation using a MACS LS column (130-042-401; Miltenyi Biotec). FLK1-expressing HB cells were then taken for flow cytometry analysis and sequencing experiments and further differentiation in blast culture into HE1, HE2, and HP populations.

To achieve this FLK1+ HB cells were plated onto gelatine-coated tissue culture flasks at a concentration of 1.6 × $10^4$ cells per $cm^2$ in Blast media (IMDM [I3390; Merck], 10% FCS, 1 x penicillin/streptomycin [P4333; Merck], 1 x L-glutamine [G7513; Merck], 0.45 mM MTG, 25 $\mu$g/ml ascorbic acid, 180 $\mu$g/ml human transferrin [T8158; Merck], 20% D4T conditioned media, 5 ng/ml VEGF [450-32; PeproTech], 10 ng/ml IL-6 [216-16; PeproTech]) and were left to incubate for 2.5 d at 37°C, 5% $CO_2$. HE1, HE2, and HP populations were FACS-sorted after incubation with KIT-APC (1:100) (553356; BD Pharmingen), TIE2-PE (1:200) (12-5987-82; eBioscience), and CD41-PECY7 (1:100) (25-0411-82; eBioscience) antibodies into HE1 (KIT+, TIE2+, CD41−), HE2 (KIT+, TIE2+, CD41+), and HP (KIT+, TIE2−, CD41+) populations.

### Enhancer reporter analysis

We performed additional next-generation sequencing on library material previously used in Edginton-White et al (2023). Libraries were resequenced on an Illumina NovaSeq 6000 using an SP 200 cycle flow cell. The sequencing data were combined with the previously published data and analysed using the published pipeline with slight modification. Briefly, reads were trimmed for quality and length using TrimGalore (Krueger et al, 2021) with parameters –nextera –length 70 –paired. Trimmed reads were aligned to the mm10 genome using Bowtie2 (v2.3.5.1) (–very-sensitive –fr –no-discordant -X 600 –no-mixed) (Doi, 1990) and the aligned reads were stringently filtered for unique high-quality alignment (mapq 40) using Samtools. The resulting bam file was converted to a bedpe file using the bamtobed function in bedtools (Quinlan & Hall, 2010) and then to a bed file by taking the first coordinate of read 1 and the final coordinate of read 2 for each pair. Duplicate fragments were then removed using the uniq function in the bed file in the shell.

The enhancer screen fragments were then further filtered, first against the negative control libraries (produced from WT cells with no enhancer reporter cassette integration) to remove PCR artefacts, then by the union of all plasmid library fragments to further remove PCR artefacts by ensuring the identical fragment was in the original plasmid libraries. Finally, the fragments were filtered for their presence in open chromatin by filtering against ATAC-Seq peaks. ATAC-Seq peaks were then assigned enhancer-positive status if they overlapped with at least 1 fragment from the enhancer screen-positive libraries. Enhancer activity levels were assigned by counting the number of high- and low-activity fragments (based on FACS sorting) at each enhancer-positive ATAC site, normalising against the total number of fragments and then assigning an activity state based on the ratio of high to low fragments at each site.

### RNA-seq library preparation

Nuclear RNA-enriched RNA-seq libraries were produced using RNA extracted with the Cytoplasmic and Nuclear RNA Purification kit (21000, 37400; Norgen) based on the manufacturer's instructions. RNA-Seq libraries were then prepared using the NEBNext Ultra II Directional RNA Library Prep Kit (E7760; NEB) total RNA with rRNA depletion protocol using 100 ng of input RNA. Libraries were sequenced using a NovaSeq 6000 SP 200 cycle flow cell.

### RNA-seq data analysis

RNA-seq reads were first filtered using RiboDetector (v0.2.7) (Deng et al, 2022)using options -t 70 -l 75 -e norrna –chunk_size 256 to remove any residual rRNA reads from the libraries. The reads were then filtered for quality and length using Trimmomatic (Bolger et al, 2014) with options LEADING:3 TRAILING:3 SLIDINGWINDOW:4:20. The trimmed reads were aligned to the mouse genome (mm10) using Hisat2 (v2.2.1) (Kim et al, 2019) using default options. Raw counts were obtained using featureCounts (Liao et al, 2014) with gene models from Ensembl as the reference transcriptome. Counts were normalized using the edgeR package (v3.40.2) (Robinson et al, 2010) in R (v4.2.3), and differential gene expression analysis was carried out using the Limma-Voom (v3.17) (Law et al, 2014) method.

## Primed enhancer identifcation

To identify primed enahncers, distal ATAC sites (>1.5 kb from a TSS) testing positive for enhancer activity were linked to genes using published promoter capture HiC data from ES cells and HPC7 cell (accession numbers: GSM2753058, runs: SRR5972842, SRR5972842, SRR5972842, SRR5972842, SRR5972842) and from HPC7 from Comoglio et al (2018) (accession numbers: GSM2702943 and GSM2702944, runs SRR5826938, SRR5826939) or where this was not possible, by closest gene, as previously published (Edginton-White et al, 2023). The enhancers were then filtered into stage transitions, for example, all enhancers occurring in both the ES and HB stages. They were then further filtered by differential gene expression looking for enhancers associated with genes that had a significant ($P$-adjusted < 0.05) twofold up-regulation across the stages with enhancer activity and were not expressed in the initial stage (CPM < 9). Gene ontology analysis of genes with primed enhancers was carried out using DAVID (Huang et al, 2009; Sherman et al, 2022).

### CRISPR/Cas9 enhancer validation

Validation of the Mecom primed enhancer by removal of the enhancer region by CRISPR/Cas9. A guide sequence was designed to target each end of the enhancer region (CACCG-GAACAGTAGCCTATCTGTCC and CACCGGGAAACCTACTGTCCCAGGA) (chr3:30353652-30354669). The guide sequences were purchased as DNA oligos and cloned into the PX458 Cs9 ad sgRNA expression vector by digestion with the BbsI restriction enzyme (NEB) and ligation using T4 Ligase (NEB). (pSpCas9(BB) – 2A-GFP (PX458) was a gift from Feng Zhang (plasmid # 48138; http://n2t.net/addgene: 48138; RRID:Addgene_48138; Addgene).

   HM-1 ES cells were transfected with the resulting vector using a Nucleofector–4D (Lonza) with the P3 Primary Cell X kit (V4XP-3024; Lonza). ES cell clones were picked and screened for successful homologous CRISPR by PCR using primers designed outside of the CRISPR guide region (AAGGCTGTCTAGCACTCGTT, GCTTTTTGCTGCTCTGCGTT). The positive clone was then confirmed by Sanger sequencing showing removal of the intended enhancer region.

### Enhancer feature comparison

To study the features associated with enhancer elements, we first performed a TF binding motif searching using Homer annotatePeaks.pl (Heinz et al, 2010) with the -m option and using the set of probability weight matrices for TFs published in Edginton-White et al (2023). To then compare the features associated with all enhancers versus primed enhancers, we created a binary matrix using pybedtools to overlap the enhancer sets with the output from the motif search and published ChIP-seq data for a number of histone modifications and TFs (GSE69101, GSE143460, GSE126496 and GSE79320). The percentage contribution of each feature to each enhancer type was then calculated and plotted.

### Network analysis

GRNs were constructed as previously described using custom Python scripts, available from Coleman et al (2023 Preprint). In brief, networks of priming elements were made using the genomic coordinates of primed enhancers at each stage transition. The positions for each TF binding motif were retrieved from these peak sets using the annotatePeaks.pl function in Homer as described in the "Enhancer Feature Comparrsion" method above. We then added coordinates for the promoters of a wide range of transcription factors and gene expression values for each differentiation stage obtained from RNA-seq experiments. These data were then used to construct a network where transcription factor genes and their target genes are represented as nodes, with the presence of a TF-binding motif in a primed enhancer targeting a gene shown as an edge. The resulting network which shows which TFs have potential to regulate which primed enhancers (shown as their associated gene) was then plotted using Cytoscape (Shannon et al, 2003).

## Data Availability

The Genome-wide data generated in this study have been deposited in the Gene Expression Omnibus database under accession codes GSE198775 and GSE244338. The genome-wide data generated in this study have also been provided as a UCSC Genome Browser Track-Hub and interactive versions of the networks are available from http://www.genomic-data.com/maytum2023.

## Supplementary Information

## Acknowledgements

This research was funded by a project grant from the Biotechnology and Biological Sciences Research Council (BBSRC) to C Bonifer and J-B Cazier. (BB/R014809/1), a BBSRC MiDTP studentship to C Bonifer for A Maytum, a BBSRC MIBTP studentship to A Maytum, and a grant from the Medical Research Council (MR/S021469/1) to C Bonifer and PN Cockerill. We thank Genomics Birmingham for expert sequencing services and the Birmingham Technology Hub and Mary Clarke for cell-sorting facilities.

### Author Contributions

A Maytum: formal analysis, investigation, and writing—review and editing.
B Edginton-White: data curation, formal analysis, validation, investigation, methodology, and writing—review and editing.
P Keane: data curation, software, formal analysis, validation, and writing—review and editing.
PN Cockerill: conceptualization, supervision, and writing—review and editing.

J-B Cazier: conceptualization, data curation, software, supervision, project administration, and writing—review and editing.

C Bonifer: conceptualization, data curation, supervision, funding acquisition, validation, project administration, and writing—review and editing.

## Conflict of Interest Statement

The authors declare that they have no conflict of interest.

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
