## [Reviewer comments · Life Science Alliance]

Life Science Alliance

Chromatin priming elements direct tissue-specific gene activity prior to hematopoietic specification

Constanze Bonifer, Alexander Maytum, Benjamin Edginton-White, Peter Keane, Peter Cockerill, and Jean-Baptiste Cazier
DOI: <https://doi.org/10.26508/lsa.202302363>

Corresponding author(s): Constanze Bonifer, University of Birmingham

Review Timeline:

Submission Date:	2023-09-11
Editorial Decision:	2023-10-25
Revision Received:	2023-11-13
Editorial Decision:	2023-11-14
Revision Received:	2023-11-15
Accepted:	2023-11-16

Transaction Report:

October 25, 2023

Re: Life Science Alliance manuscript #LSA-2023-02363-T

Prof. Constanze Bonifer
University of Birmingham
Institute for Cancer and Genomic Sciences
Institute for Biomedical Research
Birmingham, West Midlands B15 2TT
United Kingdom

Dear Dr. Bonifer,

Thank you for submitting your manuscript entitled "Chromatin priming elements direct tissue-specific gene activity prior to hematopoietic specification" to Life Science Alliance. The manuscript was assessed by expert reviewers, whose comments are appended to this letter. We invite you to submit a revised manuscript addressing the Reviewer comments.

Thank you for this interesting contribution to Life Science Alliance. We are looking forward to receiving your revised manuscript.

Sincerely,

B. MANUSCRIPT ORGANIZATION AND FORMATTING:

Reviewer #1 (Comments to the Authors (Required)):

Maytum et al. address a major issue in the relationship between cis-regulatory element action and developmental gene regulation, namely, how new cis-regulatory elements become tissue-specifically poised for activation at specific developmental transitions. Early predictions were that combinations of transcription factors alone would have enough specificity to select appropriate target genes for activation in specific cell types. However, it has become clear in the last 15 years that the effective accessibility of different genomic regions must also be regulated in a developmental hierarchy in order to constrain the patterns of transcription factor binding correctly. This paper systematically addresses how particular genomic regions become converted to accessibility, and then, in some cases, reconverted to inaccessibility. The authors use a large resource of ATAC-seq and RNA-seq data on the differentiation of hematopoietic precursors from ES cells in vitro. They also add important new data from a library of accessible DNA sequences cloned into a safe harbor expression site, showing when isolated cis-regulatory elements become functionally active as enhancers when they are divorced from their native chromatin contexts. The results provide an exceptionally rich set of examples for researchers wrestling with this question.

The authors emphasize several strong conclusions from their analyses. First, they show that it is frequent for a cis-regulatory element to become accessible in chromatin substantially before the target gene it regulates is transcriptionally active. In a large fraction of these cases, the cis-regulatory element may even be able to activate a reporter gene in the safe-harbor context at the first stage when it becomes accessible, even if it does not normally activate the native target gene until later. Second, the authors show that VEGF signalling affects a select subset of primed sites, with a specific negative effect on the opening of particular hematopoietic-specific sites. Third, they show that at least one of these primed sites is required for the correct timing of expression of the model target gene Mecom. These results and the impressively analyzed data resources assembled for this paper are an important contribution, converting the idea of chromatin priming from an anecdotal, case-by-case phenomenon to a well-documented aspect of developmental gene regulation generally. There are several places where the manuscript could be clarified.

1. With reference to Figure 3 (p. 7), the point is made that H3K27me3 is diagnostic of poised enhancers. This raises several questions.
 - a. A. Isn't H3K27me3 normally just associated with closed regulatory elements, whether poised or not? Of course in the context of elements that are H3K27me3-decorated but open, this might distinguished poised from active enhancers, but the text seems to imply that it is a sufficient criterion for poised enhancers on its own.
 - b. Are ES cells a special case? If other cells do not use H3K27me3 to mark poised enhancers, then is it that there is less H3K27me3 later overall, or less loss of H3K27me3 when the elements become active? Or is it possibly a data quality issue? Fig. 3D-F suggests that H3K27me3 (and in some cases H3K27ac) data used here are not as strong as ATAC data, but that the data for H3K27me3 seem to be better for the ES cells than for the other cell types.
2. The examples discussed in detail for the effect of VEGF are associated with hematopoietic-specific genes, which are repressed by VEGF. Accordingly, the text focuses on the regulatory elements that are closed in the presence of VEGF. But the transcription factors activated by the presence of VEGF would presumably act to open chromatin, too. Are any of these direct repressors? Please comment on the sites that are more open when VEGF is present - are any of these associated with genes that VEGF represses, or only genes that VEGF activates? If this has been established by a previous paper, please add a reference. Similarly, in the title of Fig. 4 and in the conclusion of the section on p.9, the summary of VEGF action is somewhat vaguely written. First, the data shown are examples where VEGF's continued presence only inhibits transition from primed to active - so this section seems to show more that it "restricts" expression rather than having "a role in up-regulation". Also, how many of the other enhancers show signs of direct regulation by VEGF per se, as opposed to regulation by Runx? Bcl11a, Csf2r and Csf3r are also direct targets of PU.1 in some contexts. Thus, if the conclusion of this paragraph is based on enrichment of motifs associated with VEGF signaling itself, please clarify this.
3. On p. 10 and in Fig. 5, the Mecom enhancer case is very well worked out, Fig. S6 convincingly shows that it is established in a VEGF-dependent way. However, the identification of this priming element as a determinant of gene activation timing as opposed to absolute gene activation is not shown very clearly. In the enhancer-deleted cells, Mecom expression appears to be

barely starting to rise at 64 hr, still 10-fold below the control. Are there other data that show it recovers eventually in surviving cells?

4. Minor points:

- a. With regard to Fig. 5F, are there statistical measurements of significance for these comparisons? Also, given the large effects on CD41 itself, can HE2 and HP cells be defined here without using CD41?
- b. On p. 11, with regard to the Kdr enhancer's lack of functional correlation with gene activity, could this be a more functionally effective enhancer for Kit instead? There is a cluster of Kit activity-linked cis-regulatory sites that are between the two genes but physically closer to Kdr than to Kit.
- c. There are occasional typos in the text to be fixed, e.g., "gene ontology" on p. 17.

Reviewer #2 (Comments to the Authors (Required)):

In this manuscript, the authors build on prior work using a high throughput approach to screen open chromatin for functional enhancer activity at a variety of stages from ES to the earliest production of blood progenitors. They build from this data and integrate other genomic data to now address properties of lineage-priming elements, which they define by an ATAC peak that is present in a prior stage before activation of the linked genes. They use 4 immunophenotypically defined stages of differentiation/specification (ES, Flk1+, HE1, HE2 and HP) to make these comparisons. The work differs from other analyses in the field, in that the authors can make broader conclusions due to the high throughput screening approach rather than the gene-by-gene approach. Some important conclusions are that eRNA appearance is coincident with gene activation rather than initiating prior during a priming event; that features of priming elements are generally similar to typical enhancers; that signaling events linked to developmental transitions can be mapped to particular priming elements; and that in a specific example using the Mecom locus, that deletion of an identified element can reduce/delay gene expression in a manner that impacts development, but this ultimately may be overcome through the actions of the remaining elements. The data supporting these conclusions are rigorous and build on quite a bit of prior published data to integrate for this focus on priming elements. In addition, the authors have set up a web site for others to easily explore some of their data by the larger community.

Overall the combination of the well-defined differentiation system plus high throughput, unbiased approach are unique and provide many novel insights that advance the understanding of regulatory element function, properties and interaction during differentiation. The few critiques below mainly focus on visualization/quantification of data with one point about the Mecom example which is intended to represent the function of a priming element.

Critiques

1) The points made in the text relating to Figure 2B are hard to follow from the images. I realize this diagram is represented on an interactive web version, but could the authors emphasize on the figure the points from the text (such as circle the elements of the pluripotency network they contrast between ESC and HB) or reproduce parts of the figure in zoom-in form for each of the major points? Integrating the color, density of arrows, TFs, etc. is challenging even with the figure legend. Versions of this figure that go with each of the major points in the text would be helpful. Also the names are not quite the same on both diagrams, why is that (Eg. GF1b vs GF1)? The yellow edge labels are so incredibly small it seems not worth showing them except for a zoom-in example.

2) For figure 3, it seems very important to unambiguously demonstrate that the properties of priming elements are the same or very similar to the properties of traditional enhancer elements. Therefore, the authors mention that TF enrichment is not different, but panels C and G do not clearly make that case. Some more quantitative visualization or data with a graphic of TF and TF combination enrichment should be possible comparing priming elements to all (or maybe all minus the priming elements?) that would better illustrate what I think is the point of the graphs in panel G. Specific examples:

- panel D/E/F really don't make the visual impact that H3K27me3 goes away upon differentiation-the small area around the center of the ES->HB might be argued to be less enriched, but the surrounding peaks appear to increase quite a bit-is there not a better way to quantify area under the curve of peaks in a given bin size around the center of the ATAC peak? Could this result be analyzed several orthogonal ways to illustrate the point? It would be nice to support this graphic with a p value or other quantitative measure.
- Panel G is very hard to follow (even with the figure legend) in that the H3K27me3 label is only on the first of 3 panels but it seems that this entire panel is about H3K27me3. Or is the one dot below the label the only representation of H3K27me3? If so, this should be colored or circled. It is also unclear what the individual dots represent if percentage is the axis-wouldn't percentage be a single value? The legend states that plot of proportion of features indicated in C but the features in C seem more numerous than dots-if they are, could the histone mods be color coded differently? Further explanation in the figure legend is at least needed, possibly a better visualization and quantification to support the point. If this panel represents the individual elements in panel C, could a linear fit line represent the quantitative relationship? It seems that a statistical value representing the lack of difference between enrichment at priming vs other ATAC-selected elements would make this comparison more convincing.

3) The pale color used in Figure 5C is difficult to see upon printing - the authors might consider another color scheme or bars with outline especially for the HE1 bar color

4) Similarly, the HE1 color in 5F fades away upon printing. In fact panel F may be more effective as a stack graph, since all 3 populations are present in each sample/time point and they add up to a total, which may also be relevant in this comparison. This would allow a closer comparison of the time points visually, like panels D and E.

5) the +340 enhancer of the Mecom locus is described as primed in HB and is used as the only detailed example of what happens when one deletes such a regulatory element from the endogenous gene. The issue with this example is that the ATAC seq data show that this site is very open in HE1, HE2 as well as the most active as an enhancer in the population most highly expressing Mecom transcripts (HE1). However the tiny blip in the HB population does not make this look like the best example of a fully accessible element when cells are not yet expressing Mecom. Functional data illustrate that this element was important to maintain these high levels of expression as the gene gets induced at the HB->HE1 stage. However this example is not clearly representing a priming element, as the ATAC data, YFP reporter data and expression all make it seem as simply a cell type specific enhancer element. This analysis would be more convincing if it pertained to an element that represented a clearly open chromatin site before activation of the gene (perhaps one that did not show enhancer activity) but was essential for correct timing of high-level expression. I don't know if the cellular outcome is that important for this type of analysis, but in the example shown the fact that the element seems to act simply as a traditional enhancer takes away from the focus on priming elements which seems to be a major point in this study.

Typos:

pg 4 "habour"

pg 6 "schown"

pg 7 extra comma (To examine, ...)

Dear Editor

We thank the reviewers for their insightful and constructive comments which significantly improved the paper. We have addressed all comments and hope that the paper is now suitable for publication. You will find our point-by-point response below. All changes in the text have been marked in red.

Referee 1

Maytum et al. address a major issue in the relationship between cis-regulatory element action and developmental gene regulation, namely, how new cis-regulatory elements become tissue-specifically poised for activation at specific developmental transitions. Early predictions were that combinations of transcription factors alone would have enough specificity to select appropriate target genes for activation in specific cell types. However, it has become clear in the last 15 years that the effective accessibility of different genomic regions must also be regulated in a developmental hierarchy in order to constrain the patterns of transcription factor binding correctly. This paper systematically addresses how particular genomic regions become converted to accessibility, and then, in some cases, reconverted to inaccessibility. The authors use a large resource of ATAC-seq and RNA-seq data on the differentiation of hematopoietic precursors from ES cells in vitro. They also add important new data from a library of accessible DNA sequences cloned into a safe harbor expression site, showing when isolated cis-regulatory elements become functionally active as enhancers when they are divorced from their native chromatin contexts. The results provide an exceptionally rich set of examples for researchers wrestling with this question.

Response: We thank the reviewer for their positive comments.

The authors emphasize several strong conclusions from their analyses. First, they show that it is frequent for a cis-regulatory element to become accessible in chromatin substantially before the target gene it regulates is transcriptionally active. In a large fraction of these cases, the cis-regulatory element may even be able to activate a reporter gene in the safe-harbor context at the first stage when it becomes accessible, even if it does not normally activate the native target gene until later. Second, the authors show that VEGF signalling affects a select subset of primed sites, with a specific negative effect on the opening of particular hematopoietic-specific sites. Third, they show that at least one of these primed sites is required for the correct timing of expression of the model target gene Mecom. These results and the impressively analyzed data resources assembled for this paper are an important contribution, converting the idea of chromatin priming from an anecdotal, case-by-case phenomenon to a well-documented aspect of developmental gene regulation generally.

There are several places where the manuscript could be clarified.

1. With reference to Figure 3 (p. 7), the point is made that H3K27me3 is diagnostic of poised enhancers. This raises several questions.

a. A. Isn't H3K27me3 normally just associated with closed regulatory elements, whether poised or not? Of course in the context of elements that are H3K27me3-decorated but open,

this might distinguished poised from active enhancers, but the text seems to imply that it is a sufficient criterion for poised enhancers on its own.

Response: We thank the reviewer for this insightful comment. Here we cited the wrong reference and apologise for the error. Azuara et al., described bivalent chromatin. The term “poised” enhancers which was indeed only associated with the H3K27me3 mark was originally coined by Rada-Iglesias et al., PMID: 21160473 who described such elements in ES cells in 2011. They associated them with inactive genes destined to be activated. However, in their work, open chromatin and global enhancer activity were not measured. We have altered the text accordingly. This comment prompted us to look at the K27me3 profiles more carefully and generate average profile which showed an interesting result – more in our response to referee 2 below.

b. Are ES cells a special case? If other cells do not use H3K27me3 to mark poised enhancers, then is it that there is less H3K27me3 later overall, or less loss of H3K27me3 when the elements become active? Or is it possibly a data quality issue? Fig. 3D-F suggests that H3K27me3 (and in some cases H3K27ac) data used here are not as strong as ATAC data, but that the data for H3K27me3 seem to be better for the ES cells than for the other cell types.

Response: Our work shows that this feature may indeed be a signature that is associated with open chromatin and is found only in ESCs. Our data point to a transient association of this mark on a positioned nucleosome in differentiating cells which is artificially held in place in cells where differentiation is halted. We wanted to simply report this fact without speculating too much what this would mean mechanistically as this would detract from the main message of the paper.

2. The examples discussed in detail for the effect of VEGF are associated with hematopoietic-specific genes, which are repressed by VEGF. Accordingly, the text focuses on the regulatory elements that are closed in the presence of VEGF. But the transcription factors activated by the presence of VEGF would presumably act to open chromatin, too. Are any of these direct repressors? Please comment on the sites that are more open when VEGF is present - are any of these associated with genes that VEGF represses, or only genes that VEGF activates? If this has been established by a previous paper, please add a reference.

Response: The referee is correct, the title was indeed somewhat vague and we have changed it into "VEGF signalling represses enhancer priming at hematopoietic genes" which more accurately represents what we actually show.

Similarly, in the title of Fig. 4 and in the conclusion of the section on p.9, the summary of VEGF action is somewhat vaguely written. First, the data shown are examples where VEGF's continued presence only inhibits transition from primed to active - so this section seems to show more that it "restricts" expression rather than having "a role in up-regulation". Also, how many of the other enhancers show signs of direct regulation by VEGF per se, as opposed to regulation by Runx? Bcl11a, Csf2r and Csf3r are also direct targets of PU.1 in some contexts. Thus, if the conclusion of this paragraph is based on enrichment of motifs associated with VEGF signaling itself, please clarify this.

Response: The reviewer is of course correct, up-regulation is indirect. VEGF operates by regulating the activities of several transcription factors including AP-1. In this context the reviewer is invited to look at the model depicted in our Nature Communications paper of how we think that VEGF regulates gene expression. VEGF regulates gene expression via the AP-1 / TEAD axis and up-regulates NOTCH and SOX17 which repress RUNX1. Once VEGF is gone, these factors are down-regulated, RUNX1 is upregulated, represses *NOTCH1* and *SOX17* and upregulates itself and its targets, such as *Spi1* (PU.1). We have altered the text to describe this process more precisely.

3. On p. 10 and in Fig. 5, the *Mecom* enhancer case is very well worked out, Fig. S6 convincingly shows that it is established in a VEGF-dependent way. However, the identification of this priming element as a determinant of gene activation timing as opposed to absolute gene activation is not shown very clearly. In the enhancer-deleted cells, *Mecom* expression appears to be barely starting to rise at 64 hr, still 10-fold below the control. Are there other data that show it recovers eventually in surviving cells?

Response: We apologise for this misunderstanding. *Mecom* expression does not recover, it stays low, what recovers is that over time more and more cells get over the threshold of forming HP cells – most likely by the accumulation of protein. We have altered the text to better explain this finding.

4. Minor points:

a. With regard to Fig. 5F, are there statistical measurements of significance for these comparisons?

Response: We have added the pValues into the figures

Also, given the large effects on CD41 itself, can HE2 and HP cells be defined here without using CD41?

Response: We have indeed used CD41 to define these cell populations to see how many cells pass the threshold and express enough RUNX1 to conduct the EHT. We have added an explanation to this effect in the legend.

b. On p. 11, with regard to the *Kdr* enhancer's lack of functional correlation with gene activity, could this be a more functionally effective enhancer for *Kit* instead? There is a cluster of *Kit* activity-linked cis-regulatory sites that are between the two genes but physically closer to *Kdr* than to *Kit*.

Response: We are aware of the fact that *Kit* and *Kdr* share an enhancer. However, this is not the element we are describing here which is linked to *Kdr* by CHI-C. The priming element is 100 kb closer to *Kdr*.

c. There are occasional typos in the text to be fixed, e.g., "gene ontogology" on p. 17.

Response: This typo has been corrected.

Reviewer #2:

In this manuscript, the authors build on prior work using a high throughput approach to screen open chromatin for functional enhancer activity at a variety of stages from ES to the earliest production of blood progenitors. They build from this data and integrate other genomic data to now address properties of lineage-priming elements, which they define by an ATAC peak that is present in a prior stage before activation of the linked genes. They use 4 immunophenotypically defined stages of differentiation/specification (ES, Flk1+, HE1, HE2 and HP) to make these comparisons. The work differs from other analyses in the field, in that the authors can make broader conclusions due to the high throughput screening approach rather than the gene-by-gene approach. Some important conclusions are that eRNA appearance is coincident with gene activation rather than initiating prior during a priming event; that features of priming elements are generally similar to typical enhancers; that signaling events linked to developmental transitions can be mapped to particular priming elements; and that in a specific example using the Mecom locus, that deletion of an identified element can reduce/delay gene expression in a manner that impacts development, but this ultimately may be overcome through the actions of the remaining elements. The data supporting these conclusions are rigorous and build on quite a bit of prior published data to integrate for this focus on priming elements. In addition, the authors have set up a web site for others to easily explore some of their data by the larger community.

Overall the combination of the well-defined differentiation system plus high throughput, unbiased approach are unique and provide many novel insights that advance the understanding of regulatory element function, properties and interaction during differentiation. The few critiques below mainly focus on visualization/quantification of data with one point about the Mecom example which is intended to represent the function of a priming element.

Response: We thank the reviewer for their positive comments.

Critiques

1) The points made in the text relating to Figure 2B are hard to follow from the images. I realize this diagram is represented on an interactive web version, but could the authors emphasize on the figure the points from the text (such as circle the elements of the pluripotency network they contrast between ESC and HB) or reproduce parts of the figure in zoom-in form for each of the major points? Integrating the colour, density of arrows, TFs, etc. is challenging even with the figure legend. Versions of this figure that go with each of the major points in the text would be helpful. Also the names are not quite the same on both diagrams, why is that (Eg. GF1b vs GF1)? The yellow edge labels are so incredibly small it seems not worth showing them except for a zoom-in example.

Response: We are fully aware that this is a difficult figure to understand. In essence we have several types of nodes (TF genes): (i) Genes that are not yet expressed (Yellow node colour), but primed and thus are already connected to other genes (incoming connections only) are marked in blue and by their gene name in Italics (Example: *Runx1*), (ii) TF genes that are already expressed (orange-red node colour) and the TFs leaves a footprint on other genes (incoming and outgoing connections) are marked in black letters (Example: SMAD). We use the generic name for the factor family (SMAD) if we cannot unambiguously link the factor to the connected motif. The ring of TF encoding genes is linked to a ring of target genes which are first not expressed and then expressed (change of colour from yellow to orange). We have revised the figure to (i) zoom in on a slice of the network which we actually describe, (ii) add more definitions to the actual figures and (iii) place the link to the website into the figure. We revised both the main text and the figure legend. We hope that this is clearer now.

2) For figure 3, it seems very important to unambiguously demonstrate that the properties of priming elements are the same or very similar to the properties of traditional enhancer elements. Therefore, the authors mention that TF enrichment is not different, but panels C and G do not clearly make that case. Some more quantitative visualization or data with a graphic of TF and TF combination enrichment should be possible comparing priming elements to all (or maybe all minus the priming elements?) that would could better illustrate what I think is the point of the graphs in panel G.

Response: We have tried in the text to explain better what we have actually done. We also now plotted the features representing TF motifs, TF-ChIP data and Histone modification data in different colours. To plot all enhancers minus priming elements would not make a difference as there are only a few hundred of them as compared to thousands of enhancers. In essence this plot means that priming elements use TFs and histone mods in the same way as enhancers – with the exception of ESCs.

Specific examples:

- panel D/E/F really don't make the visual impact that H3K27me3 goes away upon differentiation-the small area around the center of the ES->HB might be argued to be less enriched, but the surrounding peaks appear to increase quite a bit-is there not a better way to quantify area under the curve of peaks in a given bin size around the center of the ATAC peak? Could this result be analyzed several orthogonal ways to illustrate the point? It would be nice to support this graphic with a p value or other quantitative measure.

Response: This was actually a very important comment. To illustrate this point better, we have generated average profiles which are now plotted on top of the figures and show an intriguing phenomenon of H3K27me3 being positioned right on top of the DHS in ESCs, but flanking a complex in all of the other sites, thus showing a dip. H3K27Ac always shows a dip. There is some very interesting biology here, but as outlined above, we cannot really go into the mechanism here. However, we made comment to this effect in the text.

- Panel G is very hard to follow (even with the figure legend) in that the H3K27me3 label is only on the first of 3 panels but it seems that this entire panel is about H3K27me3. Or is the one dot below the label the only representation of H3K27me3? If so, this should be colored or circled.

Response: We are sorry for the confusion. The dots represent all the features (data-types) listed in C. The labelled dot indeed represents H3H27me3 only in ESCs. We have altered to text and the legend to explain this better.

It is also unclear what the individual dots represent if percentage is the axis-wouldn't percentage be a single value? The legend states that plot of proportion of features indicated in C but the features in C seem more numerous than dots-if they are, could the histone mods be color coded differently?

Response: We have done revised the figure accordingly.

Further explanation in the figure legend is at least needed, possibly a better visualization and quantification to support the point. If this panel represents the individual elements in panel C, could a linear fit line represent the quantitative relationship? It seems that a statistical value representing the lack of difference between enrichment at priming vs other ATAC-selected elements would make this comparison more convincing.

Response: We have added such an analysis in the figure legend.

3) The pale color used in Figure 5C is difficult to see upon printing - the authors might consider another color scheme or bars with outline especially for the HE1 bar color

4) Similarly, the HE1 color in 5F fades away upon printing. In fact panel F may be more effective as a stack graph, since all 3 populations are present in each sample/time point and they add up to a total, which may also be relevant in this comparison. This would allow a closer comparison of the time points visually, like panels D and E.

Response: We have changed the colours. We also have plotted F as a stack graph which clearly shows that the proportion of HP cells increases at late differentiation time points.

5) The +340 enhancer of the Mecom locus is described as primed in HB and is used as the only detailed example of what happens when one deletes such a regulatory element from the endogenous gene. The issue with this example is that the ATAC seq data show that this site is very open in HE1, HE2 as well as the most active as an enhancer in the population most highly expressing Mecom transcripts (HE1). However the tiny blip in the HB population does not make this look like the best example of a fully accessible element when cells are not yet expressing Mecom.

Response: We used MACS2 to call the peaks which uses a Poisson distribution to model the tag distribution and a region is considered to have a significant tag enrichment if the p-value < 10e-5 (P-value corrected for multiple comparison using Benjamini-Hochberg correction). Therefore, while the peak may look small in the screenshot, MACS2 has called it as a peak passing the threshold of significance. Note that the background is very low. Note also the ChIP data in Fig S6 showing that LMO2 and TAL1 are strongly bound to the enhancer at the HB stage (when the gene is not active) which could only be possible if there was a certain amount of open chromatin. We believe that these are the factors priming the element

Functional data illustrate that this element was important to maintain these high levels of expression as the gene gets induced at the HB->HE1 stage. However, this example is not clearly representing a priming element, as the ATAC data, YFP reporter data and expression all make it seem as simply a cell type specific enhancer element. This analysis would be

more convincing if it pertained to an element that represented a clearly open chromatin site before activation of the gene (perhaps one that did not show enhancer activity) but was essential for correct timing of high-level expression. I don't know if the cellular outcome is that important for this type of analysis, but in the example shown the fact that the element seems to act simply as a traditional enhancer takes away from the focus on priming elements which seems to be a major point in this study.

Response: All the elements we have been studying were associated with open chromatin / bound TFs prior to gene expression. This is the reason why we looked at different priming elements in Figure 4 in isolation. However, these elements are diverse. We have now highlighted this notion in the text. Some priming elements have no or little enhancer activity at any stage (i.e. *Kdr*), others show tissue-specific enhancer activity (i.e. *Hand1*) and other acquire enhancer activity later in development (*Mecom*, *Spi1*). The -340 kb enhancer does not show any activity at the HB stage in isolation but then turns into an enhancer once the right TFs come along. In any case, gene expression is massively delayed as this element drives the expression of a powerful regulator of hematopoiesis. Note that in the past (Lichtinger et al., 2012) we crippled the *Spi1* 3'URE and saw a similar delay in *Spi1* expression. What is interesting here is that some cells always manage to get over the threshold, showing again the robustness of developmental gene regulation.

Typos:

pg 4 "habour"

pg 6 "schown"

pg 7 extra comma (To examine, ...)

Response: We corrected these typos.

We hope that the reviewers are satisfied with what we have done and that the paper is now suitable for publication in Life Science Alliance.

Yours sincerely

Constanze Bonifer
(on behalf of all authors)

November 14, 2023

RE: Life Science Alliance Manuscript #LSA-2023-02363-TR

Prof. Constanze Bonifer
University of Birmingham
Institute for Cancer and Genomic Sciences
Institute for Biomedical Research
Birmingham, West Midlands B15 2TT
United Kingdom

Dear Dr. Bonifer,

Thank you for submitting your revised manuscript entitled "Chromatin priming elements direct tissue-specific gene activity prior to hematopoietic specification". We would be happy to publish your paper in Life Science Alliance pending final revisions necessary to meet our formatting guidelines.

- please consult our manuscript preparation guidelines <https://www.life-science-alliance.org/manuscript-prep> and make sure your manuscript sections are in the correct order
- please add a Category for your manuscript in our system
- please add the Twitter handle of your host institute/organization as well as your own or/and one of the authors in our system
- please upload all figure files as individual ones, including the supplementary figure files
- please include callouts for these Supplementary Figures in the manuscript - Fig S2A, Fig S3C, Fig S4C, D, Fig S6B

A. FINAL FILES:

B. MANUSCRIPT ORGANIZATION AND FORMATTING:

Sincerely,

November 16, 2023

RE: Life Science Alliance Manuscript #LSA-2023-02363-TRR

Prof. Constanze Bonifer
University of Birmingham
Institute for Cancer and Genomic Sciences
Institute for Biomedical Research
Birmingham, West Midlands B15 2TT
United Kingdom

Dear Dr. Bonifer,

Thank you for submitting your Research Article entitled "Chromatin priming elements direct tissue-specific gene activity prior to hematopoietic specification". It is a pleasure to let you know that your manuscript is now accepted for publication in Life Science Alliance. Congratulations on this interesting work.

DISTRIBUTION OF MATERIALS:

Again, congratulations on a very nice paper. I hope you found the review process to be constructive and are pleased with how the manuscript was handled editorially. We look forward to future exciting submissions from your lab.

Sincerely,
